# Disinfection Strategies for Carbapenem-Resistant *Klebsiella pneumoniae* in a Healthcare Facility

**DOI:** 10.3390/antibiotics11060736

**Published:** 2022-05-30

**Authors:** Lijia Ni, Zhixian Zhang, Rui Shen, Xiaoqiang Liu, Xuexue Li, Baiji Chen, Xiquan Wu, Hongyu Li, Xiaoying Xie, Songyin Huang

**Affiliations:** 1Department of Clinical Laboratory, Sun Yat-sen Memorial Hospital, Sun Yat-sen University, Guangzhou 510120, China; nilj@mail.sysu.edu.cn (L.N.); zhangzhx6@mail.sysu.edu.cn (Z.Z.); shenr6@mail.sysu.edu.cn (R.S.); lxqiang@mail.sysu.edu.cn (X.L.); lixx228@mail.sysu.edu.cn (X.L.); chenbj26@mail.sysu.edu.cn (B.C.); wuxiquan@mail.sysu.edu.cn (X.W.); lehyu@126.com (H.L.); 2Guangdong Provincial Key Laboratory of Malignant Tumor Epigenetics and Gene Regulation, Sun Yat-sen Memorial Hospital, Sun Yat-sen University, Guangzhou 510120, China; 3Biotherapy Center, Sun Yat-sen Memorial Hospital, Sun Yat-sen University, Guangzhou 510120, China

**Keywords:** carbapenem-resistant *Klebsiella pneumoniae*, disinfectant resistance, efflux pump

## Abstract

Disinfectant resistance is evolving into a serious problem due to the long-term and extensive use of disinfectants, which brings great challenges to hospital infection control. As a notorious multidrug-resistant bacterium, carbapenem-resistant *Klebsiella pneumoniae* (CRKP) is one of the most common and difficult pathogens of nosocomial infection. The minimum inhibitory concentration (MIC) and minimum bactericidal concentration (MBC) tests of seven kinds of disinfectants (0.1% benzalkonium bromide, 4% aqueous chlorhexidine, 75% alcohol, entoiodine II, 2% glutaraldehyde, 2000 mg/L chlorine-containing disinfectants, and 3% hydrogen peroxide) were detected by the broth dilution method. Three efflux pump genes (*oqxA*, *oqxB*, and *qacE**∆**1-sul1*) were detected by PCR. The mean MIC value of aqueous chlorhexidine from the intensive care unit (ICU) (0.0034%) was significantly higher than that from non-ICUs (0.0019%) (*p* < 0.05). The positive rates of three efflux pump genes *oqxA*, *oqxB* and *qacE**∆**1-sul1* were 60.9% (39/64), 17.2% (11/64) and 71.9% (46/64) in the detected CRKP isolates, respectively. This study discovered that CRKP strains demonstrated extensive resistance to clinical disinfectants and suggest that it is necessary to perform corresponding increases in the concentration of aqueous chlorhexidine and chlorine-containing disinfectants on the basis of current standards in the healthcare industry.

## 1. Introduction

Since the late 1980s, with the wide use of carbapenem antibiotics in the clinical setting, carbapenem-resistant Enterobacteriaceae (CRE) has been discovered and reported all over the world [1,2]. Moreover, it has become increasingly prevalent in recent years and has been considered as an urgent threat to public health [3,4,5]. Carbapenem-resistant *Klebsiella pneumoniae* (CRKP) is the most popular bacterium within the CRE family, causing longer hospital stays [1], higher medical costs [6,7], and higher mortality [8,9]. These features will add to the global estimates of the burden of antimicrobial resistance [10], which inspired us on the urgency of global action against infections [11].

The European CDC suggested that environmental cleaning, equipment reprocessing, hand hygiene, and routine surveillance were core infection prevention and control measures to minimize the risk of spread of CRE, and enhanced cleaning should be performed for areas in close proximity to CRE carriers [12]. Unfortunately, many researchers have observed resistance of multi-drug resistant bacteria to disinfectants [13]; furthermore, bacteria can acquire antibiotic resistance with the induction of disinfectants according to recent research [14]. To investigate the resistance of CRKP strains to the disinfectants commonly used in clinics, we evaluated the minimum inhibitory concentration (MIC) and minimum bactericidal concentration (MBC) of seven disinfectants to CRKP isolated from clinics in order to guide the rational use of disinfectants in clinical practice [15].

The overexpression of efflux pumps plays an important role in non-specific resistance [16,17,18,19,20]. The efflux pumps can change the structure and physiological functions of the bacterial outer membrane, and protect themselves against biocides [15,21]. Some multi-drug efflux pump families are able to mediate the efflux of multiple disinfectants and antibiotics at the same time, leading to cross resistance [22,23]. To explore the correlation of efflux pump genes with the resistance of CRKP strains to disinfectants, we detected three efflux pump genes (*oqxA**, oqxB**,* and *qacE**∆1-sul1*) [24,25,26]. *OqxA* and *oqxB* belong to the resistance-nodulation cell division (RND) family. *OqxAB* is encoded by a plasmid and can flux quinolones, tigecycline, and other antibacterial drugs, as well as quaternary ammonium compound (QAC) and biguanide disinfectants [24,25]. *qacE**∆1-sul1* belongs to the small multidrug resistance (SMR) family, and is a part of class I integrons, mediated by integrons. *q**acE**∆**1* can encode the efflux proteins to discharge QAC and biguanide disinfectants [27,28], and the *sul1* gene often causes bacterial resistance to sulfonamides [26,29]. The proteins encoded by these three efflux pump genes can flux both antibiotics and disinfectants; moreover, the multidrug efflux pumps transmitted by plasmids and integrons can promote non-specific multidrug resistance via horizontal transfer [24,25,26]. This study provides molecular epidemiological evidence for strategies to control CRKP transmission and infection in hospitals.

## 2. Methods

### 2.1. Isolation and Identification of Bacterial Strains

Our experimental strains were collected from the Sun Yat-sen Memorial Hospital (SYS Memorial Hospital) from January 2015 to December 2019. The strains were isolated from clinical patients and identified to species by the VITEK-2 automatic microorganism identifying and drug sensitivity system (bioMérieux, Marcy l’ Etoile, France) according to the manufacturer’s instructions. Susceptibility testing results were interpreted under the criteria recommended by the Clinical and Laboratory Standards Institute (CLSI, 2020). A strain of *Klebsiella pneumoniae* (*Kpn*) which was resistant to either of the carbapenem antibiotics (imipenem/ertapenem/meropenem) was considered to be a CRKP strain [1,30]. Modified carbapenem inactivation method (mCIM) and EDTA-modified carbapenem inactivation method (eCIM) tests were performed to confirm phenotypic carbapenemase production [31]. The quality control (QC) strains were *E. coli* ATCC 25922 and *Kpn* ATCC 700603, which were preserved in our laboratory.

We collected and analyzed the clinical data of CRKP strains isolated from SYS Memorial Hospital from January 2015 to December 2019 in order to provide supporting data for the use of disinfectants and the basis of research on classification of disinfectant resistance.

### 2.2. Disinfectants and Neutralizers

In this study, seven disinfectants were used, which were low-level disinfectants: 0.1% benzalkonium bromide (SYS Memorial Hospital, Guangzhou, China) and 4% aqueous chlorhexidine (SYS Memorial Hospital, Guangzhou, China); intermediate-level disinfectants: 75% ethyl alcohol (SYS Memorial Hospital, Guangzhou, China) and entoiodine II (Li Kang Disinfection Technology Co., Ltd., Shanghai, China); and high-level disinfectants: 2.0% (*w*/*v*) glutaraldehyde (Li Kang Disinfection Technology Co., Ltd., Shanghai, China), Hagrid Suli type II chlorine containing disinfectant (An Duo Fu Disinfection Technology Co., Ltd., Shenzhen, China), and 3% hydrogen peroxide (Nan Guo Pharmaceutical Co., Ltd., Guangzhou, China). The selection of neutralizers was according to the regulation of disinfection techniques in healthcare settings (WS/T 367-2012); 75% alcohol was neutralized with common nutrient broth; chlorine-containing disinfectants, entoiodine II, and hydrogen peroxide were neutralized with 0.1% sodium thiosulfate, benzalkonium bromide and chlorhexidine were neutralized with 0.3% twain 80 and 0.3% lecithin, while glutaraldehyde was neutralized with 0.3% glycine.

### 2.3. Testing the MICs and MBCs of Each Disinfectant

#### 2.3.1. The Minimum Inhibitory Concentration Test

MICs of the seven disinfectants against 64 strains of CRKP preserved in our laboratory were detected by the broth dilution method according to the guidelines of the CLSI (CLSI, 2020). Firstly, the standard bacterial concentration of 0.5 McFarland (10^8^ CFU/mL) was applied as a bacterial suspension. Secondly, double dilution of each of the disinfectants into 2.5 mL of different concentrations was performed. Then, 2.5 mL of double concentration nutrient broth was added to each concentration of disinfectants, followed by 0.1 mL of bacterial suspension, and mixed as the test group. Nutrient broth without disinfectant was inoculated with the bacteria and used as the positive control, while nutrient broth inoculated with the same volume of deionized water was used as the negative control. All the tubes were incubated at 35 °C for 48 h before the results were obtained [32,33,34,35,36]. The MICs of the disinfectants against CRKP strains demonstrated the highest dilution for aseptic growth. Experiments were performed in triplicate, with consistent results.

#### 2.3.2. The Minimum Bactericidal Concentration Test

The MBC test is a continuation of the MIC test. An amount of 0.5 mL of the sterile reaction was transferred into 4.5 mL of neutralizer specific for the particular disinfectant used in each test. The solution was fully mixed and interreacted for 10 min. An amount of 2.5 mL of the final reaction solution was added into 2.5 mL of the double concentration broth. The positive and negative control groups were prepared as described above in the MIC experiment, and neutralizer inoculated with the same volume of double concentration broth was used as the neutralizer control group. All the tubes were incubated at 35 °C for 24 h before the results were obtained [32]. The MBCs of the disinfectants against CRKP strains demonstrated the highest dilution for aseptic growth. Experiments were performed in triplicate, with consistent results.

### 2.4. PCR Detection of Efflux Pump Genes

The DNA templates of 64 CRKP strains were extracted by the boiling method. All the primers were synthesized by Guangzhou Aiji Biotechnology Co., Ltd. The primer sequences were referenced to published articles, as shown in Appendix A [26,37]. The PCR conditions were set with reference to published articles [26,37]. Amplified PCR products were analyzed on 2% agarose gel (Biowest, Nuaillé, France). The positive products of the resistance genes were confirmed using PCR followed by sequence analysis.

## 3. Results

### 3.1. Clinical Information of CRKP Strains

A total of 162 non-repetitive strains of CRKP were isolated from clinical specimens in SYS Memorial Hospital from 2015 to 2019. The average age of patients was 56.3 ± 20.7 years, among which 115 were male patients.

#### 3.1.1. The Annual Detection Amount of CRKP Strains

From January 2015 to December 2019, the detection rate of CRKP showed an overall increasing trend year by year, and there was a rapid rising peak in 2017, as shown in Figure 1A.

#### 3.1.2. Distribution of CRKP Strains in Different Specimen Types

The top four specimen types of the 162 strains of CRKP were sputum, urine, ascites/abdominal drainage fluid, and blood, accounting for 45.7% (74/162), 13.0% (21/162), 11.1% (18/162), and 9.9% (16/162), respectively, as shown in Figure 1B. The number of CRKP strains isolated from sputum was more than three times higher than that of urine.

#### 3.1.3. Distribution of CRKP Strains in Different Departments

In terms of inpatient ward distribution of CRKP strains, the top three departments were the ICU, neurology/neurosurgery, and rehabilitation departments, accounting for 41.4% (67/162), 10.5% (17/162), and 8.0% (13/162), respectively, as shown in Figure 1C. The number of CRKP strains in the ICU was much higher than in other departments and was about four times higher than that in neurology/neurosurgery, the secondary department.

#### 3.1.4. Composition of Departments That Detected CRKP Annually

From 2015 to 2019, the departments where CRKP strains were detected increased year by year. The detection number of CRKP in the ICU department was the greatest. Detailed data are shown in Appendix A.

#### 3.1.5. Distribution of Specimen Types in the Top Six Departments That Detected CRKP

We analyzed the distribution of specimen types in the top six departments, in order to identify the centralized infection sites in high-risk departments. Detailed data are shown in Appendix A.

### 3.2. Antimicrobial Sensitivity Profile

In 162 strains of CRKP, no strains were detected to be resistant to colistin or polymyxin B, while the resistance rate of tigecycline was 10% and the resistance rate of other commonly used antibiotics were all higher than 50%, as shown in Figure 1D.

### 3.3. MIC and MBC Results of CRKP Strains

#### 3.3.1. Analysis of Resistance of CRKP Strains to Seven Chemical Disinfectants Commonly Used in Clinics

The MIC and MBC results of seven clinically used chemical disinfectants against CRKP strains were expressed by diluted multiples. The higher the diluted multiple, the lower the MIC and MBC values. MIC_50_ and MIC_90_ of the CRKP strains were compared with the MIC values of the standard strains ATCC 29522 and ATCC 700603. If the diluted multiple was reduced, it indicated that the CRKP strains showed resistance to the disinfectant. The MBC data were analyzed in the same way. The results are shown in Table 1.

#### 3.3.2. Recommended Concentration for CRKP Disinfection

We referred to China’s nosocomial infection control industry standards in combination with the maximum MBC values of seven disinfectants in our study to provide recommended concentrations for CRKP disinfection. Detailed results are shown in Table 2.

#### 3.3.3. Comparisons of the MICs and MBCs of CRKP Strains Isolated from Different Wards and Different Specimens for Each Disinfectant

The results of MICs and MBCs of CRKP strains against seven clinical disinfectants were divided into two groups according to ICU ward (*n* = 23) and non-ICU wards (*n* = 41), and statistical analysis was performed by a Mann–Whitney U test of two independent samples (Figure 2). The MIC results of aqueous chlorhexidine were found to be statistically different between the ICU ward and non-ICU wards (*p* < 0.05) (Figure 2B); the MIC and MBC results of other disinfectants showed no statistical difference between the ICU ward and non-ICU wards. The results of MICs and MBCs were divided into four groups according to the different specimens, as invasive specimens (bile, catheter, ascites/abdominal drainage fluid, and blood) (*n* = 19), skin and soft tissue (*n* = 9), urine (*n* = 7) and sputum (*n* = 29), but no statistical differences between different specimen groups were found (Figure 3).

### 3.4. Analysis of Efflux Pump Genes

#### 3.4.1. Detection of Efflux Pump Genes

The detection rates of the three efflux pump genes (*oqxA*, *oqxB,* and *qacE∆1-sul1*) were 60.9% (39/64), 17.2% (11/64), and 71.9% (46/64), respectively. Partial electrophoretic results are shown in Appendix A. There were five positive detection patterns among the three efflux pump genes of the 64 strains of CRKP, among which the *qacE∆1-sul1 + oqxA* positive pattern was the dominant one, accounting for 39.1%, followed by the single gene *qacE∆1-sul1* positive pattern, accounting for 21.9%, as shown in Table 3.

#### 3.4.2. Correlation between Efflux Pump Genes and Disinfectant Resistance

The results of MIC and MBC of the disinfectants were divided into two groups according to the negative and positive genes of efflux pump, and the *t*-test of two independent samples was used for statistical analysis. There was a statistical difference in the MIC values of 0.1% benzalkonium bromide between the negative and positive *qacE∆1-sul1* gene groups (*p* < 0.05), while the MIC and MBC values of 3% hydrogen peroxide were significantly different between negative and positive *oqxA* gene groups (*p* < 0.05), and there was no statistical difference between the negative and positive efflux pump gene groups among the other disinfectants (detailed in Table 4).

## 4. Discussion

*Klebsiella pneumoniae (Kpn)* is one of the most common opportunistic pathogens, which can result in a variety of human infections, including lung, urinary tract, bloodstream, and surgical site infections [38,39]. With the emergence of carbapenem resistance, the management of *Klebsiella pneumonia* infections has been intractable and complicated [40,41]. We found that CRKP was most commonly detected in the ICU department and could be detected in all the body parts of ICU patients, mainly in sputum specimens and invasive infection specimens (ascites/abdominal drainage fluid, blood, and catheters). Hu et al. [42] and Delia et al. [43] reported that CRKP demonstrated significantly more frequent detection among ICU patients in their retrospective observational studies. This was consistent with what we observed, but in a tertiary care hospital in northern Italy, CRKP infections were concentrated in the medical ward (37.41%), geriatric ward (36.06%), and surgical ward (20.41%) [44]. Sites of infection was commonly found in the urinary tract, whereas it was the respiratory tract in our study. This indicated that CRKP infections varied greatly between different countries. In China, the isolation rate of *Kpn* in respiratory specimens has risen to the first place since 2017 [45]. In addition, our study found that the ICU department should be the primary focus of CRKP infection control works in our country, which may be related to the antibacterial drug treatment principle in the ICU ward. In China, “Heavy blow, comprehensive coverage” is the main principle utilized in the ICU ward, which refers to the elimination of all bacteria in the shortest time with the most powerful antibacterial drugs, with further descent down the ladder of treatment after the corresponding symptoms have been controlled. Such treatment could lead to the screening of “super drug-resistant bacteria” [46], including CRKP. We also found that the sharp increase in the detected number of CRKP strains in 2017 was mainly concentrated in the ICU, which should be closely related to the increase of ICU wards from two to three in our hospital in the beginning of 2017.

The high detection rate of CRKP brings challenges to nosocomial infection control.

Disinfectants play an important role in halting horizontal transmission within medical institutions [47]. However, more and more studies have observed resistance of bacteria to commonly used disinfectants [32,48,49]. In our study, we observed extensive resistance from the MIC and MBC results, which aroused our attention. Among the seven disinfectants in our experiment, six disinfectants (0.1% benzalkonium bromide, 4% aqueous chlorhexidine, entoiodine II, 2% glutaraldehyde, 2000 mg/L chlorine-containing disinfectants, and 3% hydrogen peroxide) were less sensitive to CRKP strains than the quality control strain. Chen et al. observed higher MICs and MBCs from 0.1% chlorhexidine and 0.1% povidone iodine (PVP-I) compared with the reference strains [48]. In their study, CRKP strains showed resistance to two of three disinfectants. Other research also reported that many of the CRKP strains exhibited resistance to many of the tested disinfectants [32]. Interestingly, three out of twenty-seven strains showed resistance to all of the tested disinfectants [32]. Bhatia et al. also discovered CRKP isolates showed lower susceptibility to sodium hypochlorite (4% available chlorine) in comparison to *Kpn* ATCC 700603 [49]. However, it has been speculated that the MIC and MBC levels in research were much lower than that which was used in practice, which could not accurately reflect the eliminating ability of disinfectants in real-word settings. Therefore, we compared the maximum MBC values of seven disinfectants in our study with China’s nosocomial infection control industry standards. This was because MBC is more relevant in preventing horizontal transmission of bacteria.

In clinical practice, chlorhexidine has been used as a partial preservative for over 50 years and has become the mainstay biocide in the prevention of healthcare-associated infections [50]. Jackson et al. concluded that chlorhexidine contributed to reducing the occurrence of ventilator-associated pneumonia (VAP), and recommended to implement the use of intra-oral chlorhexidine for mechanically-ventilated patients within critical care settings [51]. Chlorhexidine mouth rinse is mainly available in concentrations of 0.1%, 0.12%, or 0.2% as well as in a low concentration (≤0.06%) rinse [52]. Notably, in our study, the MBC_MAX_ of aqueous chlorhexidine against CRKP strains was 2.5 g/L (0.25%), which meant that the current concentrations of chlorhexidine mouth rinse probably could not kill all the CRKP strains colonizing the oral cavity. Skin disinfection often utilizes 4% aqueous chlorhexidine baths [53], especially in ICU patients [54]. Multiple uses of a standardized procedure of 4% chlorhexidine baths is sufficient to inhibit or kill the colonized bacteria on the skin, reducing the risk of postoperative surgical site infections [53] and reducing microbial adherence to surfaces of implantable biomedical devices [55]. According to the data in our study, the mean MIC values of aqueous chlorhexidine in the ICU ward and non-ICU ward were 0.0034% and 0.0019%, respectively, which were much lower than the current routinely used clinical concentration (4%). Therefore, the current resistance of CRKP strains to chlorhexidine only affects mucous membrane disinfection, not skin disinfection.

Based on the above discussion and the recommended concentrations in Table 2, we propose three suggestions for nosocomial infection control: (1) aqueous chlorhexidine should be ≥2.5 g/L when rinsing or gargling mucous membranes and wounds, (2) 0.1% benzalkonium bromide should avoid dilution, and (3) chlorine-containing disinfectants should be ≥1000 mg/L to eliminate known CRKP strains. These three tips should be of good help in preventing CRKP nosocomial transmission at local medical institutions.

We further explored the correlation between efflux pump genes and disinfectant resistance. The three efflux pump genes detected in our study could both efflux QAC and biguanide disinfectants [24,25,26]. Benzalkonium bromide belongs to the QAC, and chlorhexidine belongs to the biguanide disinfectants. Our study demonstrated that over 60% of the CRKP strains carried *oqxA* and *qacE∆1-sul1* genes. Only the MIC results of 0.1% benzalkonium bromide showed statistical difference between the negative and positive *qacE∆1-sul1* gene groups, and the MIC and MBC results of 3% hydrogen peroxide showed statistical difference between the negative and positive *oqxA* gene groups. The results verified the role of *qacE∆1-sul1* gene in the efflux of QAC disinfectants [27], and showed that the *oqxA* gene may be of practical significance in the efflux of hydrogen peroxide, which needs further research, but no relevant studies have been found so far. Analysis of the positive detection patterns of efflux pump genes showed that some strains carried two or three kinds of efflux pump genes at the same time, suggesting that the resistance to disinfectants may be a combination of multiple non-specific mechanisms to protect the bacteria for survival in harsh environments, which requires further investigation.

A limitation of this experiment was that the activity of efflux pumps was not detected, which was directly related with disinfectant resistance, compared with the gene carrier rates. We would like to explore the activity of the efflux pumps in the isolates with high gene carrier rates in further research.

## 5. Conclusions

The CRKP strains showed extensive resistance to clinically used disinfectants, with high efflux pump gene carrier rates. To eliminate CRKP in medical institutions effectively, the concentration of aqueous chlorhexidine and chlorine-containing disinfectants needs to be increased, and not used only on the basis of current standards. Importantly, efflux pump genes can be transmitted by plasmids or integrons, which can promote non-specific multidrug resistance via horizontal transfer. We suggest that disinfectant resistance should receive more attention and closer monitoring to guide the rational use of disinfectants and avoid the spread of the notorious multidrug-resistant bacterium.

## Figures and Tables

**Figure 1 antibiotics-11-00736-f001:**
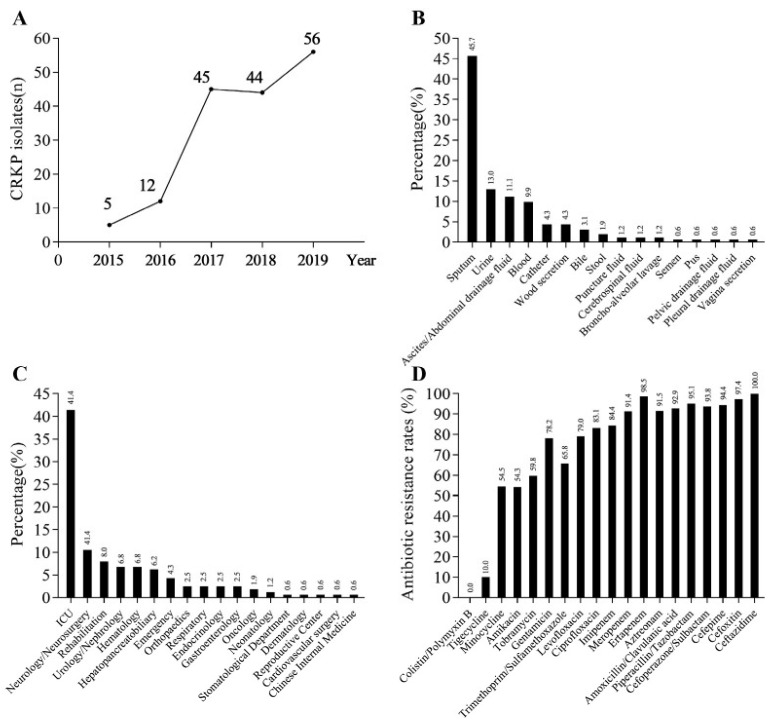
Clinical information and antimicrobial sensitivity profiles of CRKP isolates. (**A**) The detection amount of CRKP strains in SYS Memorial Hospital from 2015 to 2019. (**B**) The distribution of the 162 CRKP strains in different specimen types. (**C**) The distribution of the 162 CRKP strains in different departments; ICU: intensive care unit. (**D**) Antibiotic resistance rates of CRKP strains.

**Figure 2 antibiotics-11-00736-f002:**
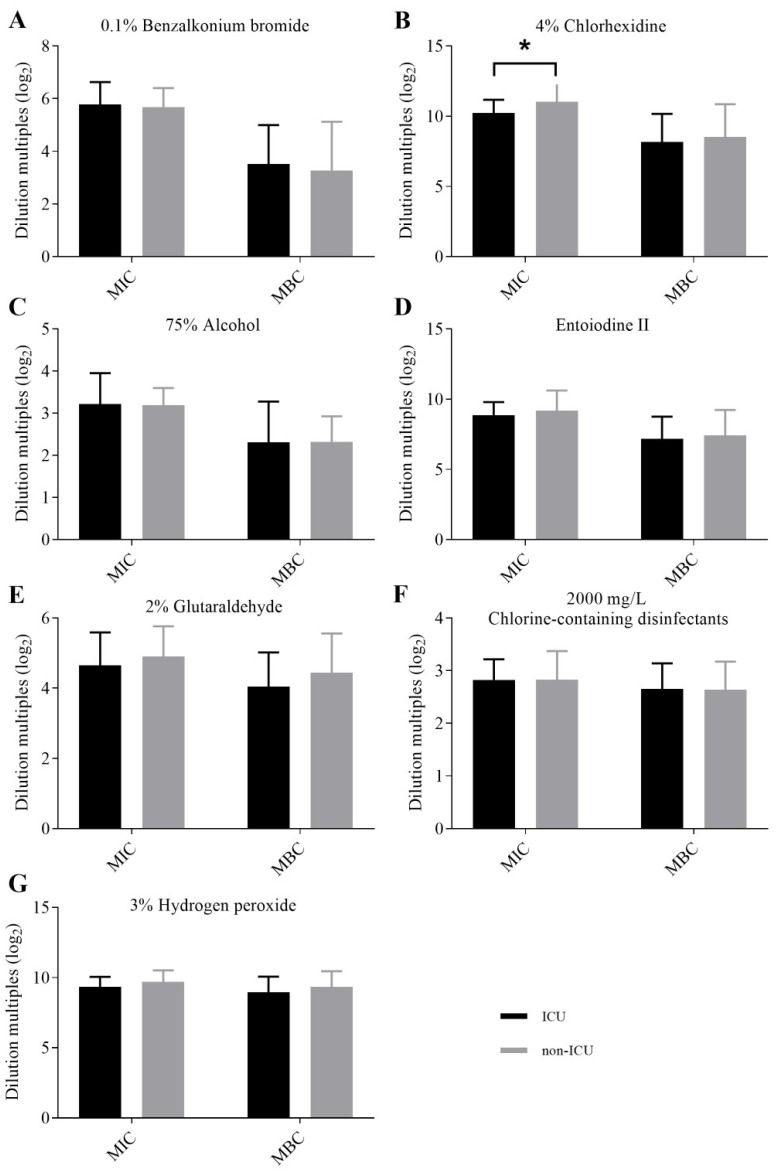
Comparisons of the MICs and MBCs of CRKP strains isolated from the ICU ward and non-ICU wards for each disinfectant. (**A**) Comparisons of the MICs and MBCs of CRKP strains isolated from ICU ward and non-ICU wards for 0.1% benzalkonium bromide. (**B**) Comparisons of the MICs and MBCs of CRKP strains isolated from ICU ward and non-ICU wards for 4% aqueous chlorhexidine. * *p* < 0.05. The MIC results of 4% aqueous chlorhexidine were statistically different between the ICU ward and non-ICU wards (**C**) Comparisons of the MICs and MBCs of CRKP strains isolated from ICU ward and non-ICU wards for 75% alcohol. (**D**) Comparisons of the MICs and MBCs of CRKP strains isolated from ICU ward and non-ICU wards for entoiodine II. (**E**) Comparisons of the MICs and MBCs of CRKP strains isolated from ICU ward and non-ICU wards for 2% glutaraldehyde. (**F**) Comparisons of the MICs and MBCs of CRKP strains isolated from ICU ward and non-ICU wards for 2000 mg/L chlorine-containing disinfectants. (**G**) Comparisons of the MICs and MBCs of CRKP strains isolated from ICU ward and non-ICU wards for 3% hydrogen peroxide. MIC: minimum inhibitory concentration; MBC: minimum bactericidal concentration; ICU: intensive care unit.

**Figure 3 antibiotics-11-00736-f003:**
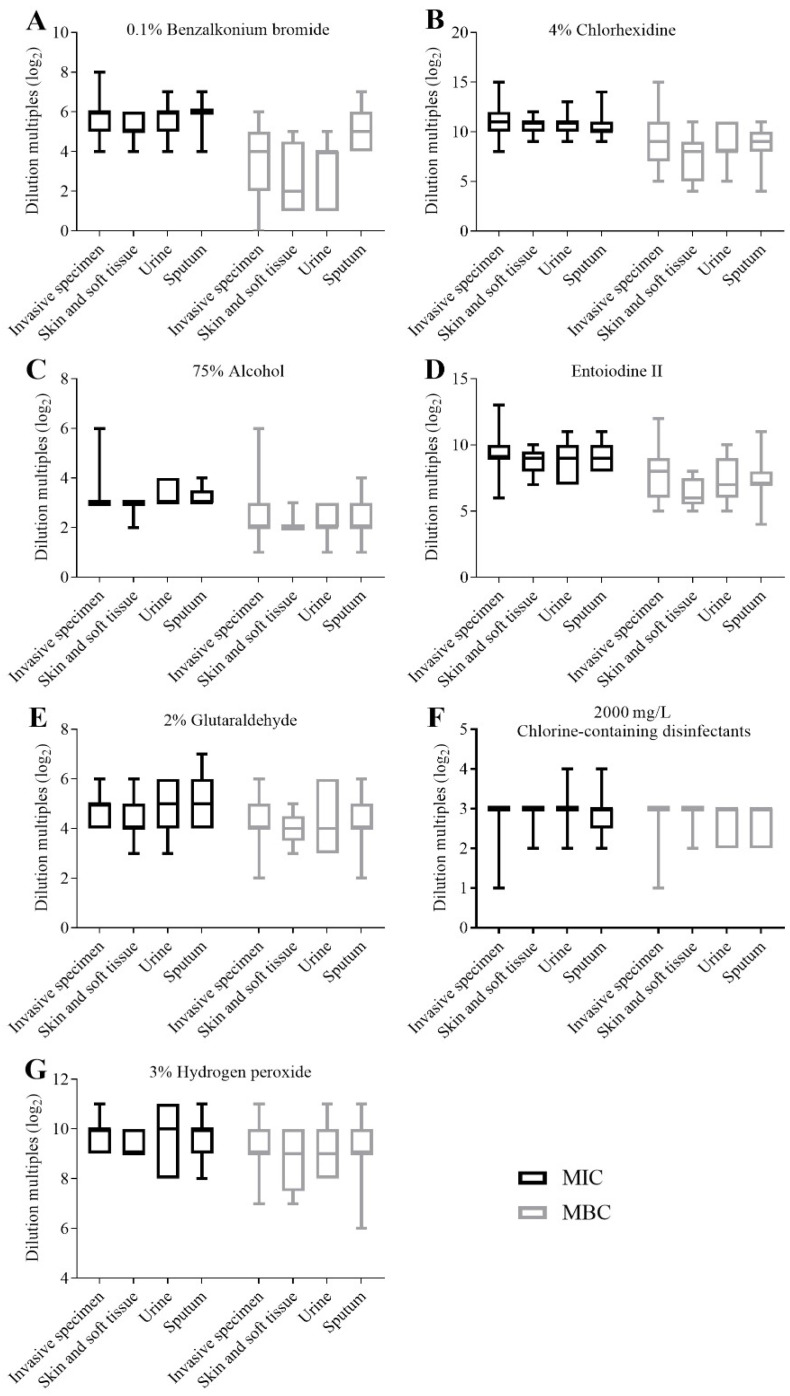
Comparisons of the MICs and MBCs of CRKP strains isolated from different specimens for each disinfectant. (**A**) Comparisons of the MICs and MBCs of CRKP strains isolated from different specimens for 0.1% benzalkonium bromide. (**B**) Comparisons of the MICs and MBCs of CRKP strains isolated from different specimens for 4% aqueous chlorhexidine. (**C**) Comparisons of the MICs and MBCs of CRKP strains isolated from different specimens for 75% alcohol. (**D**) Comparisons of the MICs and MBCs of CRKP strains isolated from different specimens for entoiodine II. (**E**) Comparisons of the MICs and MBCs of CRKP strains isolated from different specimens for 2% glutaraldehyde. (**F**) Comparisons of the MICs and MBCs of CRKP strains isolated from different specimens for 2000 mg/L chlorine-containing disinfectants. (**G**) Comparisons of the MICs and MBCs of CRKP strains isolated from different specimens for 3% hydrogen peroxide. MIC: minimum inhibitory concentration; MBC: minimum bactericidal concentration.

**Table 1 antibiotics-11-00736-t001:** MIC and MBC results of CRKP strains against seven kinds of disinfectants (dilution multiples).

Disinfectants	MICs	MIC_50_	MIC_90_	QC Strains		MBCs	MBC_50_	MBC_90_	QC Strains	
				ATCC 700603 MIC	ATCC 25922 MIC				ATCC 700603 MBC	ATCC 25922 MBC
0.1% Benzalkonium bromide	16–256	64	32	32	128	1–128	16	2	4	32
4% Aqueous chlorhexidine	256–32768	2048	512	1024	16384	16–32768	512	32	256	512
75% Alcohol	4–64	8	8	8	8	2–64	4	4	4	4
Entoiodine II	64–8192	512	256	256	16384	16–4096	128	32	128	8192
2% Glutaraldehyde	8–128	32	16	64	64	4–64	16	8	32	64
2000 mg/L Chlorine-containing disinfectants	2–16	8	4	8	8	2–8	8	4	8	8
3% Hydrogen peroxide	256–2048	1024	512	1024	2048	64–2048	512	256	512	1024

Note: MIC_50_ represents the MIC value required to inhibit the growth of 50% of the tested bacteria, and MIC_90_ represents the MIC value required to inhibit the growth of 90% of the tested bacteria. MBC_50_ represents the MBC value required to kill 50% of the tested bacteria, and MBC_90_ represents the MBC value required to kill 90% of the tested bacteria. MIC, minimum inhibitory concentration; MBC, minimum bactericidal concentration; QC, quality control; ATCC, American type culture collection.

**Table 2 antibiotics-11-00736-t002:** Recommended concentrations for CRKP disinfection according to health industry standards of the People’s Republic of China.

		Iodine	Chlorhexidine	75% Alcohol	Quaternary Ammonium Compounds	3% Hydrogen Peroxide	Chlorine-Containing Disinfectants	2% Glutaraldehyde
MBC_MAX_ ^a^		A quarter of stock solution	2.5 g/L Aqueous chlorhexidine	Half of stock solution	Stock solution (1000 mg/L)	468.75 mg/L	1000 mg/L	0.50%
Skin and mucous membrane ^b^	Skin	Stock solution	≥ 2 g/L Alcoholic chlorhexidine	Stock solution	Stock solution			
	Mucous membrane, wound		≥ 2 g/L Alcoholic chlorhexidine (wipe), ≥ 2 g/L Aqueous chlorhexidine (rinse, gargle)		1000–2000 mg/L (wipe)	3% (flush, gargle)		
Environmental surface ^c^	Bacterial propagator, mycobacterium tuberculosis, fungus, lipophilic virus			Stock solution			400–700 mg/L	
	Bacterial propagator, fungus, lipophilic virus				1000–2000 mg/L			
	All bacteria (including spores), fungus, virus					1000–2000 mg/L	2000–5000 mg/L	
Cleaning supplies ^b^	Wash towel						250 mg/L	
	Floor towel						500 mg/L	
Flexible endoscope ^d^								≥ 2% Alkaline solution
Recommended concentration ^e^			≥2.5 g/L Aqueous chlorhexidine (rinse, gargle)				1000 mg/L	

Note: ^a^ Presents maximum MBC values of seven disinfectants in our study. ^b^ Reference to regulation of disinfection technique in healthcare settings. ^c^ Reference to regulation for cleaning and disinfection management of environmental surface in healthcare settings. ^d^ Referring to regulation for cleaning and disinfection technique of flexible endoscopes. ^e^ If MBC_MAX_ is less than or equal to healthcare industry standard requirements, it is marked green; otherwise, it is marked red and listed in accordance with recommended concentrations provided by MBC_MAX_. MBC, minimum bactericidal concentration.

**Table 3 antibiotics-11-00736-t003:** Positive detection patterns of efflux pump genes.

Positive Detection Patterns	Positive Numbers (*n*)	Percentage (%)
*qacE∆1-sul1 + oqxA*	25	39.1
*qacE∆1-sul1*	14	21.9
*qacE∆1-sul1 + oqxA + oqxB*	6	9.4
*oqxA*	5	7.8
*oqxA + oqxB*	3	4.7
*qacE∆1-sul1 + oqxB*	1	1.6
*oqxB*	1	1.6

**Table 4 antibiotics-11-00736-t004:** Comparison of MIC and MBC values of disinfectants in negative and positive groups of efflux pump genes (*p* values).

Genes	Numbers	0.1% Benzalkonium Bromide		4% Aqueous Chlorhexidine		75% Alcohol		Entoiodine II		2% Glutaraldehyde		2000 mg/L Chlorine-Containing Disinfectants		3% Hydrogen Peroxide	
		MIC	MBC	MIC	MBC	MIC	MBC	MIC	MBC	MIC	MBC	MIC	MBC	MIC	MBC
*oqxA*	+: 39	0.733	0.052	0.946	0.744	0.366	0.690	0.993	0.194	0.630	0.119	0.417	0.994	0.033 *	0.001 *
	−: 25														
*oqxB*	+: 11	0.375	0.993	0.454	0.820	0.453	0.532	0.315	0.146	0.767	0.701	0.942	0.505	0.882	0.341
	−: 53														
*qacE∆1-sul1*	+: 46	0.039 *	0.807	0.285	0.679	0.861	0.820	0.160	0.885	0.093	0.498	0.968	0.413	0.367	0.563
	−: 18														

Note: * *p* < 0.05. MIC, minimum inhibitory concentration; MBC, minimum bactericidal concentration.

## Data Availability

All data generated or analyzed during this study are included in this published article. The datasets used and/or analyzed during the current study are available from the corresponding author on reasonable request.

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
