# Peer review of "Disinfection Strategies for Carbapenem-Resistant Klebsiella pneumoniae in a Healthcare Facility"

_antibiotics, 2022, doi:10.3390/antibiotics11060736_

Round 1
Reviewer 1 Report
The manuscript “Disinfection strategies of carbapenem-resistant Klebsiella pneumoniae in a healthcare facility” is devoted to studying resistance of a collection of nosocomial bacterial strains isolated from at Sun Yat-sen Memorial Hospital against seven common disinfectants. The study is certainly relevant, since the spread of antibiotic-resistant bacteria requires special attention to disinfection and hospital infection control. The emergence of resistance to disinfectants requires a revision of the protocols for their use in clinics. The authors analyzed changes in MICs and MBCs in relation to disinfectants in selected bacteria (total of 162 strains). They found that CRKP was most commonly detected in ICU department and was found also in all other departments of the hospital. According to the results, they made several suggestions for novel nosocomial infection control protocol. Of great interest is the determination of three efflux genes in samples, but no clear correlations have yet been found between their presence and resistance to disinfectants. In my opinion, the article can be accepted for publication after clarification of several issues, which are listed below.
- The introduction is too short.It is not clear why only efflux activation was chosen from a wide variety of antibiotic resistance mechanisms.Why were these three genes chosen?
- Abstract: format font size
- Line 11 Change symbol
- Line 13, 18, 36 Titles: Background, Methods, Conclusions – may be removed
- Line 127 Please provide a reference or brief details of the method.
- Line 128 It is unclear about primer sequences: they were designed by the authors?
- Line 178 How the antibiotic resistance rate was determined?
- Line 185 What is diluted multiples? It is better to analyze the MICs and MBCs directly.
- Line 189 – reformulate the sentence
- Table 1: Why are the values of MIC 90 and MBC 90 less than MIC 50 and MBC90?
- Title of Table S2 should be improved
Reviewer 2 Report
The topic of this manuscript is interesting and the findings are of significance for healthcare associated infection control and prevention. However, there are some suggestions that I have for the authors:
1. In the Method section, I would suggest to also include if carbapenem resistance confirmation tests were performed for the tested KP strains. If not, this could be introduced in the Limitations section.
2. Also in the Method section, please insert references for the MBC testing method.
3. In the Results section, the data can be presented only in the tables S2 and S3, there is no need to present the same data in text.
4. In the Discussion section, after row 259, please insert more related studies, for example DOI: 10.5603/AIT.2014.0045, DOI: 10.3390/medicina54060092, DOI: 10.1371/journal.pone.0065621, etc.
5. Overall, there is an issue with the English grammar, spelling and editing throughout the manuscript, which makes the information presented very difficult to understand (for example row 285 to 287). Also, some information presented may appear inaccurate when expressed in English language, for example row 275 to 280, row 292 to 294 etc.
Reviewer 3 Report
Overexpression of efflux pumps plays an important role in nonspecific resistance[12- 62
16]. Some multi-drug efflux pump families can mediate the efflux of multiple disinfectants 63
and antibiotics at the same time, leading to cross resistance[17, 18]. To explore the corre- 64
lation of efflux pump genes with the resistance of CRKP strains to disinfectants, we de- 65
tected three efflux pump genes (oqxA、oqxB、and qacE△1-sul1), which can flux both an- 66
tibiotics and disinfectants as reported[19-21]. This study would provides molecular epi- 67
demiological evidence for strategies to control CRKP transmission and infection in hospital- 68- Add some more background information.
Figure 2 &3- Needs to be elaborated, mainly MIC.
he CRKP strains isolated in SYS Memorial Hospital showed extensive resistance to 334
clinically used disinfectants, with high efflux pump genes carrier rates.. To eliminate 335
CRKP at local medical institutions, the concentration of aqueous chlorhexidine and chlo- 336
rine-containing disinfectants needs to be increased on the basis of current standards. Dis- 337
infectant resistance should be paid more attention and closer monitoring.
Require more explanation based on the study outcomes
Reviewer 4 Report
This study aims to provide molecular epidemiological evidence for strategies to control CRKP transmission and infection in hospitals. Your writing is confusing, the article has serious flaws. Here are some comments to improve the manuscript.
- The abstract is very long, please follow the guidelines of MDPI for the abstracts.
- The manuscript needs to be revised for English proof.
- The introduction is very brief. You need to talk more about the carbapenem-resistant Enterobacteriaceae and its economic importance. Also on many occasions, you have long sentences, please break them down to be understandable.
- Please provide references for all the information/sentences you provided in the introduction, material, methods, and discussion.
- The number of the collected samples is very low.
- What kind of samples did you collect to isolate your bacteria?
- No need for section 2.2.
- I did not see any reference for all methods you provided. Is this something you invented by yourself?
- You need to provide a table of materials and methods to distribute your samples.
- I wonder, why do not you run the antimicrobial resistance assay in 96 well plates not in test tubes
- the screened number of the bacteria is very high.
- The study design to detect MBC is not correct, please check other publications.
- The work done in this paper is not enough for publication.
Round 2
Reviewer 4 Report
Thank you for providing explanation. There is still minor grammatical errors